# Meta-Analysis Examining the Importance of Creatine Ingestion Strategies on Lean Tissue Mass and Strength in Older Adults

**DOI:** 10.3390/nu13061912

**Published:** 2021-06-02

**Authors:** Scott C. Forbes, Darren G. Candow, Sergej M. Ostojic, Michael D. Roberts, Philip D. Chilibeck

**Affiliations:** 1Department of Physical Education Studies, Faculty of Education, Brandon University, Brandon, MB R7A6A9, Canada; 2Faculty of Kinesiology and Health Studies, University of Regina, Regina, SK S4SOA2, Canada; Darren.candow@uregina.ca; 3Applied Bioenergetics Lab, Faculty of Sport and Physical Education, University of Novi Sad, Lovcenska 16, 21000 Novi Sad, Serbia; sergej.ostojic@chess.edu.rs; 4School of Kinesiology, Auburn University, Auburn, AL 36849, USA; mdr0024@auburn.edu; 5College of Kinesiology, University of Saskatchewan, Saskatoon, SK S7N 5B2, Canada; phil.chilibeck@usask.ca

**Keywords:** supplements, hypertrophy, sarcopenia

## Abstract

Creatine supplementation in conjunction with resistance training (RT) augments gains in lean tissue mass and strength in aging adults; however, there is a large amount of heterogeneity between individual studies that may be related to creatine ingestion strategies. Therefore, the purpose of this review was to (1) perform updated meta-analyses comparing creatine vs. placebo (independent of dosage and frequency of ingestion) during a resistance training program on measures of lean tissue mass and strength, (2) perform meta-analyses examining the effects of different creatine dosing strategies (lower: ≤5 g/day and higher: >5 g/day), with and without a creatine-loading phase (≥20 g/day for 5–7 days), and (3) perform meta-analyses determining whether creatine supplementation only on resistance training days influences measures of lean tissue mass and strength. Overall, creatine (independent of dosing strategy) augments lean tissue mass and strength increase from RT vs. placebo. Subanalyses showed that creatine-loading followed by lower-dose creatine (≤5 g/day) increased chest press strength vs. placebo. Higher-dose creatine (>5 g/day), with and without a creatine-loading phase, produced significant gains in leg press strength vs. placebo. However, when studies involving a creatine-loading phase were excluded from the analyses, creatine had no greater effect on chest press or leg press strength vs. placebo. Finally, creatine supplementation only on resistance training days significantly increased measures of lean tissue mass and strength vs. placebo.

## 1. Introduction

The age-related decrease in lean tissue mass and strength are two main factors that contribute to the development of sarcopenia [1]. Approximately 10% of the adult population ≥60 years of age has sarcopenia [2], which has a profound negative effect on functional independence and overall quality of life [3]. Furthermore, sarcopenia is associated with other age-related diseases and health conditions such as osteoporosis and physical frailty [3,4]. Several lines of research suggest that sarcopenia is caused by age-related changes in muscle protein kinetics, neuromuscular function and physiology, skeletal muscle morphology, inflammation, and mitochondrial dysregulation [1,5,6]. In addition to these cellular and mechanistic changes, insufficient physical activity and nutritional intake also contribute to sarcopenia [3,7]. Interestingly, dietary intake of creatine, a key component for muscular bioenergetics, decreases with age [8].

The combination of creatine supplementation and resistance training has the potential to serve as an effective countermeasure to the age-related loss in lean tissue mass and strength, possibly by influencing anaerobic energy metabolism, calcium and glycogen regulation, muscle protein kinetics, inflammation and oxidative stress [3,9,10]. However, results from individual studies (*n* = 20) are mixed, with 10 studies showing beneficial effects on measures of lean tissue mass and/or strength (leg press, chest press) while 10 studies found no greater benefit from creatine vs. placebo (Table 1). While numerous methodological variables may explain these inconsistent findings, differences in creatine dosage and frequency of ingestion during the resistance training program is likely involved [10]. For example, of the 20 studies performed, 6 studies used a lower-dose creatine strategy (≤5 g/day for 12–26 weeks), 6 studies used a creatine-loading phase (≥20 g/day for 5–7 days) followed by a lower-dose creatine strategy (≤5 g/day) while 2 studies used a creatine-loading phase (≥20 g/day for 5–7 days), followed by a higher-dose creatine strategy (>5 g/day for 11 weeks). Furthermore, 6 studies used a higher-dose creatine strategy (>5 g/day) for 8–52 weeks. Finally, 4 of the 20 studies had participants ingest creatine only on resistance training days. The average sample size across studies was only 34 participants. Therefore, these studies were likely unpowered to detect small differences in lean tissue mass and strength (leg press, chest press). To overcome low statistical power across studies, meta-analyses are often performed. 

To date, three meta-analyses have been performed involving creatine supplementation and resistance training in older adults [9,32,33]. Collectively, results showed that creatine and resistance training increased measures of lean tissue mass by ~1.2 kg and strength (leg press, chest press) more than placebo and resistance training. However, no sub-analyses were performed to determine whether the dosage of creatine used or the frequency of ingestion (i.e., only on resistance training days) influenced measures of lean tissue mass and/or strength. Since the publication of these meta-analyses, two additional studies involving creatine supplementation and resistance training in older adults have been published. Therefore, the purpose of this review was to (1) perform updated meta-analyses comparing creatine vs. placebo (independent of dosage and frequency of ingestion) during a resistance training program on measures of lean tissue mass and strength, (2) perform meta-analyses examining the effects of different creatine dosing strategies (lower: ≤5 g/day vs. higher: >5 g/day), with and without a creatine-loading phase (20 g/day for 5–7 days, and (3) perform meta-analyses determining whether creatine supplementation only on resistance training days influences measures of lean tissue mass and strength. Results from these meta-analyses may provide important information for the design of optimal creatine supplementation strategies for older adults. 

## 2. Materials and Methods

We have previously published two meta-analyses in 2014 [9] and 2017 [32]. Based on our expertise in the literature, we updated these meta-analyses with recently published studies since the date of the 2017 publication [32]. PubMed and SPORTDiscus databases were searched. Similar to our previous meta-analysis [32] key terms and similar phrases were used (creatine OR creatine monohydrate OR creatine supplementation OR creatine-loading) AND (weight lifting OR weight training OR resistance training, OR resistance exercise OR strength training) AND (age OR middle-age OR older adults OR elderly). Studies with the following criteria were included: (1) healthy and chronic disease participants with a mean age >50 years of age; (2) must be a randomized control trial (RCT) where participants were randomized to an intervention group consisting of creatine monohydrate with resistance training or placebo with resistance training; (3) included outcome measures of whole-body lean tissue mass (determined with dual-energy X-ray absorptiometry [DEXA], hydrostatic weighing, air displacement plethysmography, bioelectrical impedance, or multi-site ultrasound), or upper-(chest press) or lower-body (leg press) muscular strength. Studies were excluded if they were <5 weeks in duration. 

Two researchers (S.C.F. and D.G.C.) determined whether the relevant articles were to be included, and any disagreements were resolved by consensus. Databases were searched up until February 2021. Means and standard deviations for baseline and post-training measurements were extracted from each study for estimation of mean changes and the standard deviation of mean changes across the interventions. Change scores were calculated as the pre-training mean subtracted from the post-training mean. Standard deviations (SD) for the change scores were estimated from pre and post-training standard deviations (SD-pre and SD-post) using the following equation derived from the *Cochrane Handbook for Systematic Reviews of Interventions*: SD change score = [(SD pre)^2^ + (SD post)^2^ − 2 * (correlation between pre and post scores) * SD pre * SD post]^1/2^

We used 0.8 as the assumed correlation between pre- and post-scores. Heterogeneity was evaluated using χ^2^ and I^2^ tests where heterogeneity was indicated by either χ^2^
*p*-value ≤ 0.1 or I^2^ test value > 75%. We used a fixed-effects model for our meta-analysis. Weighted mean differences were calculated for lean tissue mass, along with the 95% CI. As units of measurement differed across studies for measurements of strength, calculated standardized mean differences (SMDs) and 95% CIs for leg press and chest press strength were used. Forest plots were generated using Review Manager 5.3 Software (Cochrane Community, London, UK). Significance was established at *p* ≤ 0.05. Funnel plots were generated and inspected for publication bias. Adverse events were also extracted. 

### Sub-Analyses

To examine the influence of creatine dosage, dosing strategy was extracted and classified as either higher (>5 g/day) or lower (≤5 g/day), as well as whether the study included a “loading phase” (≥20 g/day for 5–7 days) and whether creatine was only consumed on resistance training days. Only two studies [15,17] used a creatine dosage <5 g/day. Both absolute and relative (based on body mass) dosing strategy studies were included. We estimated an absolute dose of creatine ingested per day from the product of the average body mass and the relative dose. Several sub-analyses were performed to examine the effects of creatine within each classification. Furthermore, sensitivity analysis was conducted to explore whether the overall effects depended on a single specific study. 

## 3. Results

### 3.1. Lean Tissue Mass

The analysis of 16 RCTs with 18 treatment arms (*n* = 509) revealed that creatine supplementation and resistance training increased measures of lean tissue mass vs. placebo and resistance training (Figure 1: mean difference = 1.32 kg [95% CI: 0.93, 1.72] *p* < 0.000001). 

Sub-analyses showed that higher-dose creatine, with and without a creatine-loading phase, produced significant gains in lean tissue mass vs. placebo (Figure 1: mean difference = 1.21 kg [95% CI: 0.57, 1.85] *p* = 0.0002). Even when studies incorporating a creatine-loading phase were excluded, higher-dose creatine remained effective (Appendix A: mean difference = 1.16 kg [95% CI: 0.49, 1.82] *p* = 0.0006). 

Lower-dose creatine, with and without a creatine-loading phase, increased lean tissue mass vs. placebo (Figure 1: mean difference = 1.40 kg [95% CI: 0.89, 1.91] *p* < 0.00001). When studies incorporating a creatine-loading phase were excluded, lower-dose creatine was still more beneficial than placebo (Appendix A: mean difference = 1.81 kg [95% CI: 1.20, 2.42] *p* < 0.00001). 

### 3.2. Chest Press Strength

The analysis of 17 RCTs with 19 treatment arms (n = 456) revealed that creatine supplementation and resistance training significantly increased chest press strength vs. placebo and resistance training (Figure 2: standard mean difference = 0.28 [95% CI: 0.09, 0.47] *p* = 0.004). 

Subanalyses showed that studies using higher-dose creatine, with and without a creatine-loading phase, found similar effects compared to the placebo (Figure 2 and Appendix A; *p* > 0.05). However, sensitivity analysis indicated that omitting the Candow et al. [27] study changed the overall effect to significantly favor creatine (Appendix A; *p* = 0.008). 

Studies using a creatine-loading phase followed by lower-dose creatine revealed a significant benefit in favor of creatine (Figure 2: standard mean difference = 0.33 [95% CI: 0.05, 0.61] *p* = 0.02). However, when studies incorporating a creatine-loading phase were excluded from the analysis, lower-dose creatine was similar to the placebo (Figure 3; *p* = 0.12). 

### 3.3. Leg Press Strength 

The analysis of 15 RCTs with 17 treatment arms (n = 426) revealed that creatine supplementation and resistance training significantly increased leg press strength vs. placebo and resistance training (Figure 4: standard mean difference = 0.20 [95% CI: 0.00, 0.39] *p* = 0.05). 

Sub-analyses showed that higher-dose creatine, with and without a creatine-loading phase, produced greater gains in leg press strength vs. placebo (Figure 4: mean difference = 0.29 [95% CI: 0.04, 0.54] *p* = 0.02). However, when studies incorporating a creatine-loading phase were excluded, higher-dose creatine was similar to the placebo (Figure 5: *p* = 0.12). 

Studies using lower-dose creatine, with and without a creatine-loading phase, had no greater effect on leg press strength vs. placebo (Figure 4; *p* = 0.69 and Appendix A; *p* = 0.88). 

### 3.4. Creatine Only on Training Days

When only including studies that provided creatine on resistance training days, there were significant overall effects for favoring creatine on measures of lean tissue mass (Figure 6: mean difference = 1.73 kg [95% CI: 0.87, 2.89] *p* < 0.0001), chest press strength (Figure 7: standard mean difference = 0.58 [95% CI: 0.20, 0.96] *p* = 0.003), and leg press strength (Figure 8: standard mean difference = 0.44 [95% CI: 0.06, 0.81] *p* = 0.02). Of note, Cooke et al. [30] incorporated a creatine-loading phase followed by lower-dose creatine (≤5 g/day) whereas the studies by Candow et al. [25,26] used higher-dose creatine (>5 g/day). 

### 3.5. Publication Bias

Funnel plots for each meta-analysis were visually inspected and showed no evidence of publication bias.

### 3.6. Adverse Events

In the lower-dose studies (≤5 g/day), 10 studies reported no adverse events. One study reported a single mild bout of gastro-intestinal distress from creatine [16] and one study reported an overuse shoulder injury following creatine supplementation [18]. Neither of these studies used a loading phase. 

In the higher-dose studies (>5 g/day), five studies reported no adverse events. Two studies similarly reported five incidences of gastrointestinal distress from creatine and two incidences from placebo and two incidences of muscle cramps from both the creatine and placebo group [27,28]. One of the two studies utilizing a loading phase reported an increase in GI distress during the loading phase [29]. 

## 4. Discussion

The most important results from these meta-analyses were: (1) creatine supplementation (independent of creatine-loading, maintenance dosage and frequency of ingestion) during a resistance training program increased measures of lean tissue mass and strength compared to the placebo and resistance training in older adults, (2) the combination of creatine-loading followed by lower-dose creatine (≤5 g/day) was effective for increasing chest press strength, (3) the combination of creatine-loading and higher-dose creatine (>5 g/day) was effective for increasing leg press strength, (4) creatine supplementation only on resistance training days significantly increased measures of lean tissue mass and strength compared to the placebo. These results have application for the design of effective creatine supplementation strategies for older adults. For example, older adults wanting to improve whole-body lean tissue mass and strength may expect these benefits from creatine supplementation (i.e., ≥5 g) either daily or only on training days during a resistance training program.

Increasing whole-body lean tissue mass and strength is fundamental for mitigating sarcopenia and associated conditions of osteoporosis and physical frailty *(3)*. Older adults specifically looking to improve upper-body strength (perhaps to improve functionality, posture and/or the ability to perform upper-body activities of daily living such as carrying groceries) may need to load with creatine before proceeding to a lower daily dosage (≤5 g) during their resistance training program. To specifically increase lower-body strength (perhaps to improve balance, reduce the risk of falls and/or the ability to perform lower-body activities of daily living such as climbing stairs), older adults may need to load with creatine before proceeding to a higher daily dosage (>5 g) during their resistance training program. While some have hypothesized creatine may have harmful effects [34], a plethora of evidence shows no adverse events (compared to the placebo) with long-term supplementation [35,36,37].

Previous meta-analyses have shown greater gains in measures of lean tissue mass (~1.2–1.3 kg) and strength from creatine supplementation and resistance training in older adults compared to the placebo [9,32,33]. Since the date of these publications, two additional studies [24,27] have been performed. When these studies were included in the current meta-analyses, creatine supplementation and resistance training still increased measures of lean tissue mass (~1.32 kg) and strength compared to the placebo. Collectively, results across meta-analyses suggest that the combination of creatine supplementation and resistance training has the potential to mitigate sarcopenia. Although none of the studies included in any of the meta-analyses were powdered to directly examine the effects of creatine vs. placebo in older adults diagnosed with sarcopenia, sub-analyses from three studies showed that the combination of creatine and resistance training eliminated the classification of sarcopenia in 11 older adults [20,23,26]. Creatine supplementation may augment lean tissue mass and strength through various mechanisms [3,4,10,32,37]. First, supplementation increases intramuscular PCr resulting in greater resynthesis of ATP during and following muscle contractions. Supplementation also increases muscle GLUT-4 content and translocation to the sarcolemma which may increase glucose uptake and subsequent glycogen resynthesis [38,39]. Creatine supplementation facilitates calcium re-uptake via creatine kinase into the sarcoplasmic reticulum, and this may increase myofibrillar cross-bride cycling, cell swelling, the expression of myogenic transcription factors (i.e., Mrf4, myogenin), satellite cell proliferation, and the expression of growth factors (i.e., insulin-like growth factor-1) [40,41]. Creatine supplementation enhances the activation of protein kinases downstream in the mammalian target of rapamycin (mTOR) pathway, and this may subsequently reduce measures of muscle protein catabolism (i.e., leucine oxidation, urinary 3-methylhistidine) [25,31]. Finally, creatine supplementation could reduce inflammation (i.e., cytokines) [42,43] and oxidative stress [44,45,46], and again, this may help reduce the loss of lean tissue mass with aging [4].

Incorporating a creatine-loading phase during the initial stages of a resistance training program was determined to be important for improving upper- and lower-body strength. It is well established that creatine-loading results in significant elevations in intramuscular creatine levels [47]. However, the magnitude of the effect on strength outcome measures may also depend on the maintenance dosage of creatine used for the remainder of the training program.

Regarding upper-body strength, older adults who loaded with creatine and then proceeded to ingest lower-dose creatine daily experienced greater upper body strength gains compared to those on placebo. However, independent of a creatine-loading phase, lower-dose creatine supplementation was no more effective than placebo. When all studies were included in the analysis, higher-dose creatine supplementation daily, with and without a creatine-loading phase, had no greater effect on upper-body strength compared to the placebo. However, sensitivity analysis showed that when the Candow et al. [27] study was removed, results became significant in favor of creatine. In this study, older males supplemented with higher-dose creatine daily during supervised, whole-body resistance training for 52 weeks. Results showed that changes in upper-body strength were similar between creatine and placebo over time. Both creatine and placebo groups experienced large increases in strength over time (creatine: ~69 kg; placebo: ~76 kg) which likely masked any effect from creatine supplementation. 

Regarding lower-body strength, creatine-loading followed by higher-dose creatine daily had a favorable effect on strength whereas creatine-loading followed by lower-dose creatine daily had no greater effect compared to the placebo. The magnitude of responsiveness to creatine supplementation in older adults may depend on initial intramuscular creatine levels [10,48]. There is some evidence to suggest that phosphocreatine stores decrease with aging [10], especially in muscles of the lower limbs, possibly due to type-II muscle fiber atrophy, reduced participation in high-intensity activities and reduced meat consumption [32]. Furthermore, lower-body muscle groups are more negatively affected (i.e., greater strength deficit) by the aging process than upper-body muscle groups [49]. Therefore, to overcome possible age-related changes in muscle creatine content and lower-body muscle morphology, higher creatine dosages (as opposed to lower-creatine dosages) may be needed on a daily basis after a creatine-loading phase to improve lower-body strength in older adults. 

Most importantly, all the studies identified as using a high dose (i.e., >5 g/day) were based on a relative dosing strategy (based on body mass; g/kg/day), while all the low dose studies used an absolute dosing strategy (g/day). As such, future research is required to directly compare an absolute and relative strategy to determine which method is superior. 

Older adults who ingested creatine only on resistance training days experienced greater gains in measures of lean tissue mass and strength compared to the placebo. One study implemented a creatine-loading phase prior to lower-dose creatine daily [30] whereas the other studies implemented a higher-dose daily strategy [25,26]. A common theme across all studies was that creatine was consumed within 60 min’ post-exercise. While the mechanistic actions of creatine were not determined in these studies, previous research has shown that prior muscle contractions (i.e., resistance training sessions) stimulate greater creatine uptake into muscle [50] possibly through increased activation of creatine transport kinetics [51,52]. These results may be important, as compliance to a creatine supplementation program may be higher when creatine is only consumed on training days. However, it is unknown whether older adults experience the same muscle benefits when consuming creatine supplementation daily vs. only on training days during a resistance training program. In addition, a provision of creatine from a regular diet should be accounted for a total exposure to creatine in this population since creatine consumption varies in the elderly [53].

Although the focus of this review was on combining creatine with resistance exercise, there appears to be some benefits of creatine without concomitant exercise in older adults [54,55]. Future research may be warranted to examine the dose of creatine to enhance muscle performance without exercise. 

## 5. Conclusions

Increasing whole-body lean tissue mass and strength is fundamental for mitigating sarcopenia and associated conditions of osteoporosis and physical frailty [3]. Similar to previous meta-analyses [9,32], our results showed that creatine supplementation and resistance training increases measures of lean tissue mass and strength in older adults vs. placebo. However, unique and important results from our sub-analyses indicate that a creatine-loading phase is important for older adults wanting to improve muscle strength. In addition to a creatine-loading phase, a lower daily dosage of creatine (≤5 g) appears sufficient to improve upper-body strength. However, a higher daily dosage of creatine (>5 g) after the loading phase is needed to increase lower-body strength. Regarding the effects of creatine ingestion frequency, creatine supplementation only on resistance training days significantly increased measures of lean tissue mass and strength compared to placebo. 

## Figures and Tables

**Figure 1 nutrients-13-01912-f001:**
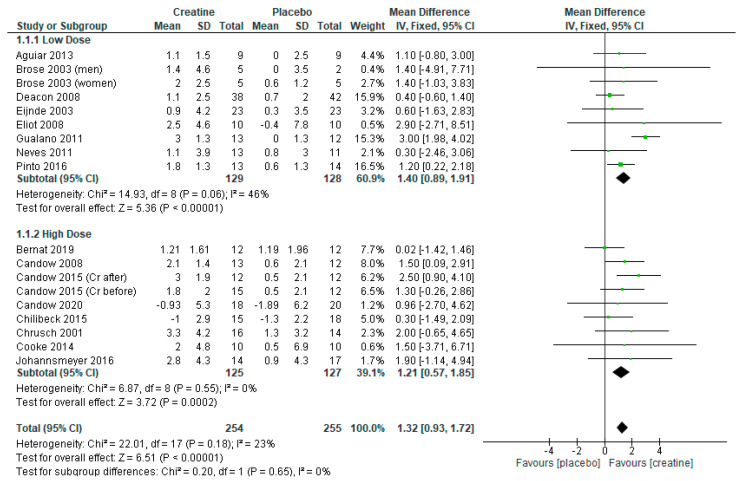
Forest plot of studies on lean tissue mass with sub-analyses using lower-dose creatine studies (≤5 g/day) and of higher-dose creatine studies (>5 g/day) on lean tissue mass.

**Figure 2 nutrients-13-01912-f002:**
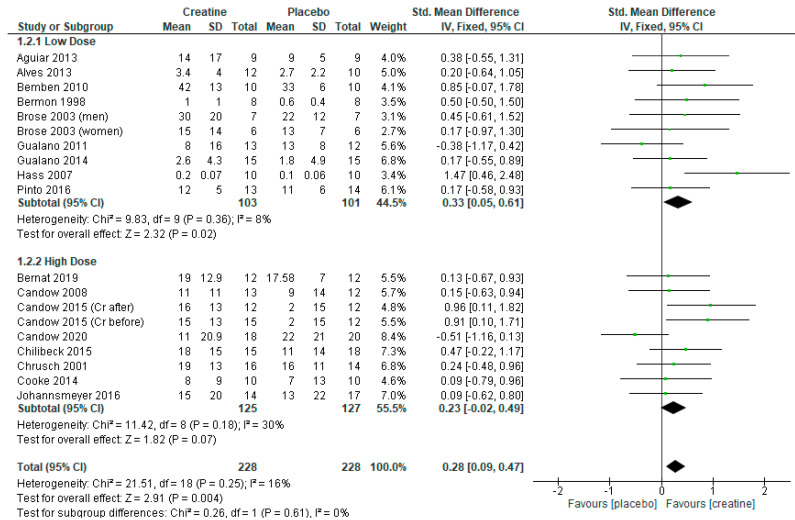
Forest plot of studies on chest press strength.

**Figure 3 nutrients-13-01912-f003:**
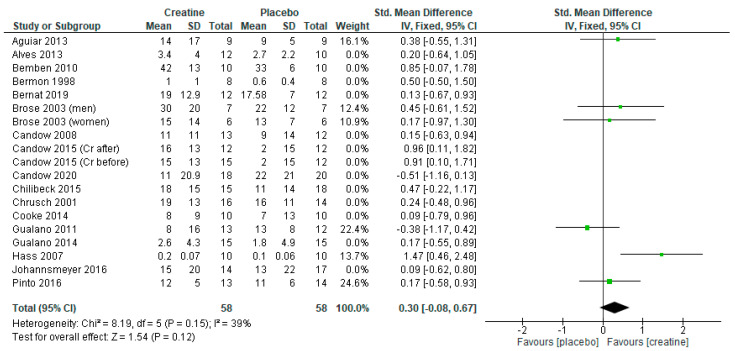
Forest plot of lower-dose creatine studies (≤5 g/day) on chest press strength with exclusion of creatine loading studies.

**Figure 4 nutrients-13-01912-f004:**
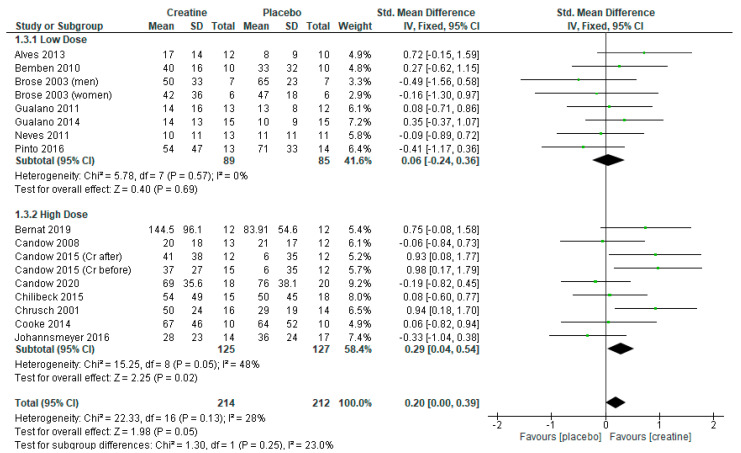
Forest plot of studies on leg press strength.

**Figure 5 nutrients-13-01912-f005:**
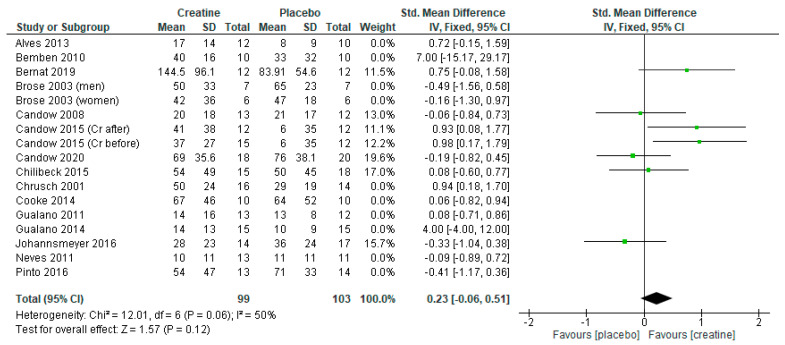
Forest plot of higher-dose creatine studies (>5 g/day) on leg press strength with exclusion of creatine-loading studies.

**Figure 6 nutrients-13-01912-f006:**
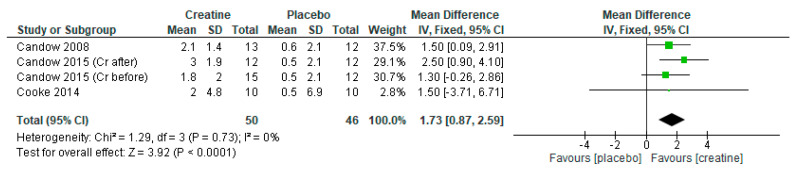
Forest plot of studies on lean tissue mass.

**Figure 7 nutrients-13-01912-f007:**
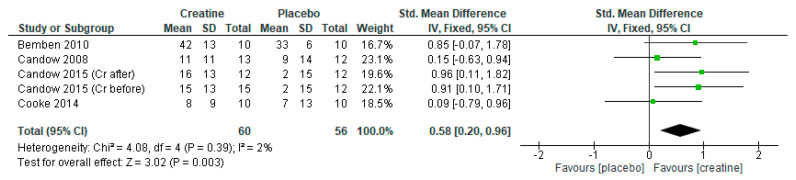
Forest plot of studies on chest press strength.

**Figure 8 nutrients-13-01912-f008:**
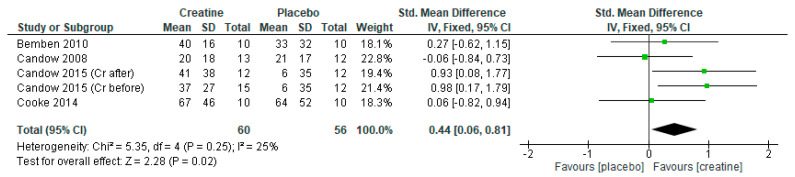
Forest plot of studies on leg press strength.

**Table 1 nutrients-13-01912-t001:** Study characteristics, dosing strategy, and outcomes of research examining the influence of creatine in older adults with a resistance training program.

First Author, Year	Population	Supplement Protocol		Resistance Training	Duration	Outcomes
		Loading Protocol	Maintenance Dose			
Lower-Dose/Absolute Studies (≤5 g/day)						
Alves et al. [11]	*N* = 47; healthy women, Mean age = 66.8 years (range: 60–80 years)	CR 20 g/day for 5 days	CR (5 g/day) or PLA	RT = 2 days/wk	24 wks	↔ 1RM strength compared to RT + PLA
Aguiar et al. [12]	*N* = 18; healthy women; Mean age = 65 years	None	CR (5 g/day) or PLA	RT = 3 days/wk	12 wks	CR ↑ gains in fat-free mass (+3.2%), muscle mass (+2.8%), 1RM bench press, knee extension, and biceps curl compared to PLA
Bemben et al. and Eliot et al. [13,14]	*N* = 42; healthy men; age = 48–72 years	None	CR (5 g/day)	RT = 3 days/wk	14 wks	↔ lean tissue mass, 1RM strength
Bermon et al. [15]	*N* = 32 (16 men, 16 women); healthy; age = 67–80 years	CR 20 g/day for 5 days	CR (3 g/day) or PLA	RT = 3 days/wk	7.4 wks (52 days)	↔ lower limb muscular volume, 1-, 12-repetitions maxima, and the isometric intermittent endurance
Brose et al. [16]	*N* = 28 (15 men, 13 women); healthy; age: men = 68.7, women = 70.8 years	None	CR (5 g/day) or PLA	RT = 3 days/wk	14 wks	CR ↑ gains in lean tissue mass and isometric knee extension strength; ↔ type 1, 2a, 2x muscle fiber area
Deacon et al. [17]	*N* = 80 (50 men, 30 women); COPD; age = 68.2 years	CR 22 g/day for 5 days	CR (3.76 g/day) or PLA	RT = 3 days/wk	7 wks	↔ lean tissue mass or muscle strength
Eijnde et al. [18]	*N* = 46; healthy men; age = 55–75 years	None	CR (5 g/day) or PLA	Cardiorespiratory + RT = 2–3 days/wk	26 wks	↔ lean tissue mass or isometric maximal strength
Gualano et al. [19]	*N* = 25 (9 men, 16 women); type 2 diabetes; age = 57 years	None	CR (5 g/day) or PLA	RT = 3 days/wk	12 wks	↔ lean tissue mass
Gualano et al. [20]	*N* = 30; “vulnerable” women; Mean age = 65.4 years	CR 20 g/day for 5 days	CR (5 g/day) or PLA	RT = 2 days/wk	24 wks	CR + RT ↑ gains in 1RM bench press and appendicular lean mass compared to PLA + RT
Hass et al. [21]	*N* = 20 (17 men, 3 women with idiopathetic Parkinson’s disease); Mean age = 62 years	CR 20 g/day for 5 days	CR (5 g/day) or PLA	RT = 2 days/wk	12 wks	CR ↑ chest press strength, chair rise performance; ↔ Leg extension 1RM, muscular endurance
Neves et al. [22]	*N* = 24 (postmenopausal women with knee osteoarthritis); Age = 55–65 years	CR 20 g/day for 1 week	CR 5 (g/day) or PLA	RT=3 days/wk	12 wks	CR ↑ gains in limb lean mass. ↔ 1RM leg press
Pinto et al. [23]	*N* = 27 (men and women); healthy; age = 60–80 years	None	CR (5 g/day) or PLA	RT = 3 days/wk	12 wks	CR ↑ gains in lean tissue mass. ↔ 10 RM bench press or leg press strength
Higher-Dose/Relative Studies (>5 g/day)						
Bernat et al. [24]	*N* = 24 healthy men; age = 59 ± 6 years	None	CR (0.1 g/kg/day; ~9.5 g/day) or PLA	High-velocity RT = 2 days/wk	8 wks	↔ muscle thickness, physical performance, upper body muscle strength. CR ↑ leg press strength, total lower body strength
Candow et al. [25]	*N* = 35; healthy men; age = 59–77 years	None	CR (0.1 g/kg/day; ~8.6 g/day) or PLA	RT = 3 days/wk	10 wks	CR ↑ muscle thickness compared to PLA. CR ↑ 1RM bench press ↔ 1RM leg press
Candow et al. [26]	*N* = 39 (17 men, 22 women); healthy; age = 50–71 years	None	CR (0.1 g/kg; ~7.7 g/day) before RT, CR (0.1 g/kg; ~8.8 g/day) after RT, or PLA	RT = 3 days/wk	32 wks	CR after RT ↑ lean tissue mass, 1RM leg press, 1RM chest press compared to PLA
Candow et al. [27]	*N* = 38; healthy men; age = 49–67 years	None	CR (On training days: 0.05 g/kg before and 0.05 g/kg after exercise; total ~9.3 g/day) + 0.1 g/kg/day on non-training days (2 equal doses)	RT = 3 days/wk	12 months	↔ lean tissue mass, muscle thickness, or muscle strength
Chilibeck et al. [28]	*N* = 33; healthy women; Mean age = 57 years	None	CR (0.1 g/kg/day; ~6.9 g/day) or PLA	RT = 3 days/wk	52 wks	↔ lean tissue mass and muscle thickness gains between groups. ↑ relative bench press strength compared to PLA.
Chrusch et al. [29]	*N* = 30; healthy men; age = 60–84 years	CR 0.3 g/kg/d for 5 days	CR 0.07 g/kg/day; ~6.2 g/day or PLA	RT = 3 days/wk	12 wks	CR ↑ gains in lean tissue mass. CR ↑ 1RM leg press, 1RM knee extension, leg press endurance, and knee extension endurance. ↔ 1RM bench press or bench press endurance.
Cooke et al. [30]	*N* = 20; healthy men; age = 55–70 years	CR 20 g/day for 7 days	CR 0.1 g/kg/day or ~8.8 g/day on training days	RT = 3 days/wk	12 wks	↔ lean tissue mass, 1RM bench press, 1RM leg press
Johannsmeyer et al. [31]	*N* = 31 (17 men, 14 women); healthy; age = 58 years	None	CR 0.1 g/kg/day; ~7.8 g/day or PLA	RT = 3 days/wk	12 wks	CR ↑ gains in lean tissue mass and 1RM strength in men only

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
