# Peer review of "Meta-Analysis Examining the Importance of Creatine Ingestion Strategies on Lean Tissue Mass and Strength in Older Adults"

_nutrients, 2021, doi:10.3390/nu13061912_

Round 1
Reviewer 1 Report
Overall, it is a well written paper and the topic could be important for practitioners who are aiming to improve training outcomes for aging population. My main concern with the paper is the analysis on the dose response.
In this analysis, it suggests that Cr dosage is important to the upper (>5g) and lower (<5g) strength. However, it appears that the high dose studies were taking the body weight (0.1g to 0.05g/kg/day) into consideration when determining the amount of Cr. For low-dose studies, they were given an absolute amount of Cr. I think it would be more appropriate to use relative Cr dosage as a cutoff point. Here is my concern, from most of the high dose studies, normally 0.1g/kg/day was used. In a low dose study (women average 60kg; men average 80kg), then for women, it was ~0.08g/kg/day and for men, it was ~0.06g/kg/day. This will definitely result in bias analysis. Especially the Candow study that was removed, they used 0.05g/kg/day and found no change in strength. If the cutoff of high and low dose is using g/kg/day, this study will probably fall under the low dose study. Therefore, it is important to go back to the low-dose study and estimate the g/kg/day Cr intake.
In the discussion, it mentions that type II muscle fiber atrophy can be a factor that explains the low and high dose. Based on this, it would be important to separate sex and disease state when performing the analyses.
Author Response
Thank you very much for your time and effort, particularly during the COVID-19 pandemic. We have responded in point-by-point fashion.
Reviewer 1
Overall, it is a well written paper and the topic could be important for practitioners who are aiming to improve training outcomes for aging population. My main concern with the paper is the analysis on the dose response.
RESPONSE: Thanks for the comment. We have responded to the comment below regarding the analysis and how we classified each paper. We have also added further information in the manuscript for clarity.
In this analysis, it suggests that Cr dosage is important to the upper (>5g) and lower (<5g) strength. However, it appears that the high dose studies were taking the body weight (0.1g to 0.05g/kg/day) into consideration when determining the amount of Cr. For low-dose studies, they were given an absolute amount of Cr. I think it would be more appropriate to use relative Cr dosage as a cutoff point. Here is my concern, from most of the high dose studies, normally 0.1g/kg/day was used. In a low dose study (women average 60kg; men average 80kg), then for women, it was ~0.08g/kg/day and for men, it was ~0.06g/kg/day. This will definitely result in bias analysis. Especially the Candow study that was removed, they used 0.05g/kg/day and found no change in strength. If the cutoff of high and low dose is using g/kg/day, this study will probably fall under the low dose study. Therefore, it is important to go back to the low-dose study and estimate the g/kg/day Cr intake.
Response: We completely agree that since all the high dose studies used a relative dosing strategy and the low dose studies used an absolute dosing strategy it is unknown whether it is the impact of relative vs. absolute or the total grams of creatine ingested. We have clarified this both in the methods (how we classified studies) and discussion section (as a potential limitation that warrants further research).
However, we need to clarify that the Candow et al. 2020 used 0.1 g/kg/day total (they gave 0.05 g/kg before AND 0.05 g/kg after training). We have made this clearer in the manuscript (particularly in the table). We have also indicated the estimated absolute grams for each of the high dose studies in the Table for comparisons.
In the discussion, it mentions that type II muscle fiber atrophy can be a factor that explains the low and high dose. Based on this, it would be important to separate sex and disease state when performing the analyses.
Response: First, it is speculation that type II muscle fibers may alter the response. Syrotuik and Bell examined responders and non-responders (using the same dose) and found responders had a higher number of Type II muscle fibers. Secondly, most studies used both males and females. There was only 1 study that used female only participants in the high dose/relative dosing strategy and four that used a low dose strategy. As such, making comparisons between genders and dosing strategies is limited. Further, there were three studies that included diseased states (Deacon et al. used COPD; Hass et al. Parkinsons; Neves et al. knee osteoarthritis), on all analysis we removed individual studies and the results were not significantly changed.
Reviewer 2 Report
The paper by Drs Forbes and colleagues on the importance of creatine supplementation for older adults containing a number of full-fletched meta-analyses is researched and compiled very carefully, the manuscript is written very well and is convincingly supported by corresponding tables. The topic is highly relevant since creatine supplementation is moving away from sports only and is increasingly engaged into the realm of health care and prevention. This is especially relevant for seniors, elderly and frail persons, as well as for people in rehabilitation.
The collated data support the notion that creatine supplementation, especially if used in combination with exercise is indeed effective for gaining lean muscle mass and improve muscle strength. These data are highly relevant for maintaining and improving quality of life of senior persons.
I have only a few remarks and questions that should possible be addressed by the authors.
1) The authors compiled data from studies describing either creatine supplementation by seniors tgether with resistance training, or alternatively, creatine supplementation taken only on the days of resistance training and exercise. Question: do the authors have information on whether creatine supplementation without resistance training at all has any positive health effects? Even if this should not be the case, it would deserve some discussion and possibly adding some references.
2) The authors do not discriminate of whether creatine supplementation took place before, during or after excercise and resistance training. As there is some published evidence, although not very highly-pitched that creatine may be more effective if taken after training, this should be mentioned and discussed. see : Moon-A et al. 2013; https://pubmed.ncbi.nlm.nih.gov/24304199/
minor: on line 246 a small correction may be needed in saying that creatine kinase associated to the sarcoplasmic reticulum is facilitating via PCr the reuptake of calciium (it is not creatine per se) and a corresponding reference should be added.
Author Response
The paper by Drs Forbes and colleagues on the importance of creatine supplementation for older adults containing a number of full-fletched meta-analyses is researched and compiled very carefully, the manuscript is written very well and is convincingly supported by corresponding tables. The topic is highly relevant since creatine supplementation is moving away from sports only and is increasingly engaged into the realm of health care and prevention. This is especially relevant for seniors, elderly and frail persons, as well as for people in rehabilitation.
Response: Thank you very much for your time and effort.
The collated data support the notion that creatine supplementation, especially if used in combination with exercise is indeed effective for gaining lean muscle mass and improve muscle strength. These data are highly relevant for maintaining and improving quality of life of senior persons.
I have only a few remarks and questions that should possible be addressed by the authors.
1) The authors compiled data from studies describing either creatine supplementation by seniors tgether with resistance training, or alternatively, creatine supplementation taken only on the days of resistance training and exercise. Question: do the authors have information on whether creatine supplementation without resistance training at all has any positive health effects? Even if this should not be the case, it would deserve some discussion and possibly adding some references.
Response: Thank you very much for your comment. We have recently published a review examining creatine without exercise (https://pubmed.ncbi.nlm.nih.gov/33502271/) which found similar results to the Moon et al 2013 review. Both reviews conclude that there are some benefits of creatine without exercise, however it appears to be less effective as when combined with resistance exercise. We have mentioned this in the discussion. “Although the focus of this review was on combining creatine with resistance exercise, there appears to be some benefits of creatine without concomitant exercise in older adults (55, 56). Future research may be warranted to examine the dose of creatine to enhance muscle performance without exercise.”
2) The authors do not discriminate of whether creatine supplementation took place before, during or after excercise and resistance training. As there is some published evidence, although not very highly-pitched that creatine may be more effective if taken after training, this should be mentioned and discussed. see : Moon-A et al. 2013; https://pubmed.ncbi.nlm.nih.gov/24304199/
Response: Thank you very much for your comment. We have previously examined the effect of creatine timing (i.e., before, during, or after training) and we have data (accepted for publication) that suggests that creatine timing (before vs. after) has no effect on muscle thickness or strength adaptations in young adults and we have previously published data in older adults that suggests that timing (before vs. after) in older participants had no effect on muscle adaptations (PMID: 25993883). Further, we also recently published a study examining creatine DURING training and found that this is a viable strategy (this study was done in younger adults (PMID: 32599716). As such, we believe based on our current and previous data that creatine timing in close proximity to training is not a major factor influencing the findings.
minor: on line 246 a small correction may be needed in saying that creatine kinase associated to the sarcoplasmic reticulum is facilitating via PCr the reuptake of calciium (it is not creatine per se) and a corresponding reference should be added.
Response: We have revised accordingly.